# Exploring the perceptions of the effect of the COVID-19 pandemic on the mental well-being and medical education of medical students in Northern Ireland, in addition to the perceived barriers to seeking support; a qualitative study

Claire Whiteside [1,2]*, Gonnie Klabbers [1]

1 Department of Health Ethics and Society, Faculty of Health Medicine and Life Sciences Maastricht University, Maastricht, The Netherlands, 2 School of Medicine, Dentistry and Biomedical Sciences, Queens University Belfast, Belfast, Northern Ireland

* cwhiteside05@qub.ac.uk

## Abstract

### Introduction

The COVID-19 pandemic had a negative effect on population mental health. Medical students may have been particularly affected, whom prevalence of mental health conditions was already high before the pandemic hit, due to the difficult and stressful academic programme. In Northern Ireland specifically, mental well-being levels are the lowest across the UK; however limited research exists examining the medical student cohort. This study explores Northern Irish medical students' perceptions on how the pandemic affected their mental health, their progress within medical education and perceived barriers to accessing support services in Northern Ireland.

### Methods

A qualitative study of phenomenological design involving 15 in-depth semi-structured interviews. The interviews were conducted amongst individuals who were 1st-4th year medical students when the pandemic was officially declared in Northern Ireland in March 2020. The interviews were transcribed, and thematic analysis was carried out using NVivo V12 qualitative data analysis software.

### Results

Results demonstrated the COVID-19 pandemic had a considerable negative impact on participants' mental health; a variety of interlinked social, individual and/or psychological and organisational factors led to increased levels of stress, anxiety and depression. This had a secondary negative impact on participants' medical education progress through reducing motivation, causing burnout and impostor syndrome. Unexpectedly; there were some

**Data Availability Statement:** The data cannot be shared publicly because of initial agreements made with the Maastricht University ethics committee upon obtaining ethical clearance, and agreements with participants upon obtaining consent. Participants were informed all data would be anonymized and only available to the principal researcher and supervising researcher. All relevant qualitative data quotations are contained within the manuscript. All data is in the form of recordings which contain identifiable and sensitive information, and are securely stored by the principal researcher. The qualitative data is available upon request from Claire Whiteside, principal researcher, and from Maastricht University Ethical Committee via email (ethical_clearance_gh@maastrichtuniversity.nl), for researchers who meet the criteria for access to confidential data.

**Funding:** The author(s) received no specific funding for this work.

**Competing interests:** The authors have declared that no competing interests exist.

perceived positive outcomes, including improved appreciation for work-life balance and resilience. Participants reported various barriers to seeking help amongst this difficult time period; also categorizable into social, individual and/or psychological and organisational factors, for example; stigmatisation, fear and perfectionistic tendencies.

## Discussion and conclusion

There is a pressing demand for heightened support availability, personally tailored mental health assistance and an effort to reduce mental health stigma in Northern Ireland. This study highlights the complex multifactorial nature of mental health. Medical schools must provide additional services to protect well-being during particularly challenging periods and dismantle the barriers preventing individuals from accessing vital support.

## 1. Introduction

The WHO demonstrates the pandemic had a significant impact triggering and exaggerating various mental health diseases globally, particularly among health care professionals (HCPs) and medical students [1, 2]. COVID-19 forced sudden changes to medical education; including online learning, clinical placement suspensions and sudden uptake of hospital employment to assist with shortages [3, 4]. Pre-pandemic, the 2019 Global Burden of Disease Study demonstrated mental health disorders are amongst the 10 leading global causes of disease burden; responsible for 80.8 million disability adjusted life years [5]. Pre-pandemic literature demonstrates medical students have a lower baseline mental health compared to the general population; one in three globally having anxiety [6]. This may be because medical education is a very challenging degree; emotionally and academically, therefore it puts a strain on students' well-being, often resulting in anxiety, depression and sleep disruptions [6–8]. Moreover, support services are often not utilised or available [9]. In Northern Ireland (NI) specifically, pre-pandemic research demonstrates mental health diseases are 20–25% higher compared to the rest of the United Kingdom. Particularly conditions including anxiety, depression, substance misuse and PTSD; this disparity is believed to be secondary to the 30 year-long civil unrest period, 'The Troubles' [10–13].

Poor mental well-being can have detrimental direct and indirect impacts on work, education and physical health, potentially leading to psychological distress, anxiety, post-traumatic stress disorder (PTSD) and suicidal behaviours. Therefore, medical students' mental health is a significant growing global health concern as it lays the foundations for future professional mental health stability and high-quality patient care. With suicide being the 2nd leading mortality cause amongst the 15–24 age category in the United States, suicidal tendencies are a particular concern amongst medical students; the ideation often continuing or increasing as they transition into professionals [14–16]. When left untreated, HCPs with mental health problems can lead to low quality patient-doctor relationships, medical errors, economic strain and fatalities [17, 18]. A 2019 British Medical Association (BMA) survey concluded 80% of HCPs were at high burnout risk [19]. The WHO recognises mental well-being as an integral health component; nevertheless, a mental health stigma exists amongst HCPs and medical students which limits disclosure of personal mental health problems [1, 9]. This is due to the invisibility of mental illness, and fear that seeking help will affect their fitness to practice [20–22]. This

stigma limits discussion and utilisation of mental health support amongst HCPs and students; causing them to seek support only prior to a major crisis; leading to poorer outcomes [23].

At the start of the COVID-19 pandemic in early 2020, NI experienced a significant rise in infection rates, similar to global trends. Initial cases were confirmed in March, and rapid viral spread quickly followed; resulting in 2,546 COVID-related deaths between 1st March 2020 and 31st January 2021; 15.8% of all deaths. The majority of these deaths were in the 80–89 age category [24]. By April 2020, NI had implemented strict lockdown measures in response to surging rates, with the number of confirmed cases reaching thousands. This escalation highlighted the urgent need for public health interventions and ongoing monitoring [25]. During this period there were significant changes to medical education in NI; final year students were employed as doctors prematurely to assist in the pandemic front line, 3rd/ 4th years were employed as medical student technicians (MST) or health care assistants, and 1st/2nd years were shifted to completely online education [26–28]. The MST role involved being enlisted in a diverse range of clinical environments including emergency departments, wards or testing centres; supporting with responsibilities such as venepuncture, cannulation, electrocardiograms, personal care and manual handling [28]. Various medical students assisted with the mass vaccine distribution which began on 8th December 2020; this assisted in controlling infection rates; by February 2021 27% of the population had received their 1st dose [29, 30]. However, despite vaccines being readily available, in October 2021 the government reported NI to have the lowest uptake of COVID-19 vaccines in the UK [31]. Numerous quantitative studies in various settings in NI have evidenced the psychological impact of the pandemic on medical students and HCPs, including negative outcomes such as loneliness, alcohol misuse, domestic violence, financial difficulties, employment and educational changes [1, 32, 33]. However, there are gaps in the knowledge understanding how these identified factors are intricately linked to students' mental health. Additional qualitative research is required to gain an in-depth understanding into medical students' experiences, thoughts and perceptions on how their mental health was affected during the pandemic, in addition to how the consequences influenced their medical education progress in NI. The conceptualisations underlying the present study are based on the work of Htay and colleagues [34] identifying the triggering and relieving factors linked to the impact of COVID-19 on psychological well-being amongst HCPs, and by the work of Stuijfzand and colleagues, who divided the factors that determine the effect of COVID-19 into 4 dimensions; organisational, social, personal and psychological [35].

This research will allow for evaluation of medical students' perceptions on how their mental health was affected by the pandemic. The aim is to understand how to increase mental health support utilisation and reduce mental health stigma amongst medical students through increasing mental health discussion. Upon completing the study; these aims were achieved. Gaining this knowledge provides an opportunity to increase future pandemic preparedness and accessibility of mental health support services within NI. Overall, this research contributes to improving future medical student mental well-being, therefore consequently improving HCP mental well-being. The specific study objectives include:

1. Determine the current junior doctors' and medical students' perceptions of the effect of COVID-19 on mental health during medical school.

2. Determine the perceived triggering and relieving factors affecting medical students' mental health during the COVID-19 pandemic.

3. Identify the perceived barriers amongst medical students in accessing mental health support services throughout medical school.

4.  Determine junior doctors' and medical students' perceptions on how COVID-19 effect on their mental health had a secondary impact on their medical education experience.

## 2. Methods

### Study design

A qualitative research approach involving one-to-one interviews was utilised, allowing the principal researcher to gain an in-depth understanding into participants' perceptions on the COVID-19 effect on their mental well-being during medical school. A phenomenological method was used; this is a type of qualitative research which aims to understand and provide an explanation for existing phenomena. It involved studying participants' emotions, perceptions and beliefs surrounding their lived experiences during the pandemic, while simultaneously exploring how the individuals understand such experiences [36]. The interviews were carried out in a semi-structured manner and continued until data saturation [37]. Interview topics covered factors affecting participants' mental health during the pandemic, perceptions of mental health stigma, barriers to support uptake, and impact on medical education. The participant recruitment and data collection process took 1 month in total, while data analysis and research write-up took an additional 2 months.

### Study population and sampling methods

15 interviews were carried out amongst foundation year one (FY1) and two (FY2) doctors who were practising for either the Health and Social Care Northern Ireland (HSCNI) or NHS England, in addition to 4th year, 5th year and intercalating medical students. All participants were from NI, and/or went to medical school in NI during the pandemic. There were 12 females and 3 males with an age range of 22–26 years and average age 23.8 years. Participant demographics are demonstrated in Table 1. Intercalating referred to students who had completed medicine year 3 and were taking a year break before returning to year 4. These participants collectively were 1st-4th year medical students during the start of the pandemic in

**Table 1. Participant demographics.**

| Participant Number | Current level | Age | Gender | Length of interview / minutes |
|---|---|---|---|---|
| 1 | Intercalating Medical student | 22 | Female | 30 |
| 2 | Foundation Year 2 Doctor (in NI) | 26 | Female | 21 |
| 3 | Intercalating Medical Student | 22 | Male | 34 |
| 4 | Intercalating Medical Student | 22 | Female | 26 |
| 5 | Foundation Year 2 Doctor (in NI) | 26 | Female | 31 |
| 6 | Foundation Year 1 Doctor (in England) | 25 | Female | 29 |
| 7 | 5th Year Medical Student | 26 | Female | 33 |
| 8 | Intercalating Medical Student | 22 | Female | 32 |
| 9 | Intercalating Medical Student | 22 | Female | 31 |
| 10 | Foundation Year 1 Doctor (in NI) | 25 | Female | 32 |
| 11 | Foundation Year 1 Doctor (in England) | 25 | Female | 34 |
| 12 | 5th Year Medical Student | 24 | Female | 27 |
| 13 | 4th Year Medical Student | 22 | Male | 41 |
| 14 | 4th Year Medical Student | 22 | Female | 29 |
| 15 | Foundation Year 2 Doctor (in NI) | 26 | Male | 26 |

March 2020. Utilising individuals from four different medical education levels allowed for increased sampling variation and higher perspective range. Participants on sick leave from university/work, or actively seeking treatment for a current mental health condition were excluded to avoid exacerbating mental health difficulties in vulnerable individuals. The participants were recruited through purposeful sampling and snowball sampling; initial participants were approached within the principal researcher's own network of contacts within NI, then additional participants recruited through existing participants [38, 39]. Participant recruitment took place from May 24th, 2023, until June 29th, 2023. Ethical approval was obtained from Maastricht University under registration number FHML/GH_2023.024 on 23rd May 2023, in addition to written approval from the NI medical school Dean.

## Data collection

Before commencing each interview, the study was appropriately explained; verbally and through a participant information sheet and recruitment poster. Written and informed consent were obtained from the research participants, and any participant questions answered prior to commencing each interview. No minors were included in the study. There was a safety protocol in place; in the event of a concerning disclosure, the interview would be discarded, and the participant would be directed to relevant occupational health services. The data were anonymised and securely stored to ensure participant privacy; it will be stored for 10 years post publication in accordance with Maastricht University Ethical Code of conduct.

## Data analysis

With participant consent, the interviews were audio recorded using a separate offline device, transcribed verbatim, anonymised and later reviewed for accuracy. The transcripts were analysed using a combination of deductive and inductive methods by the principal researcher, in parallel with the interviews to identify recurring themes, adjust interview questions and confirm when no new themes arose. A coding scheme was developed using the NVivo V.12 software. With guidance from Htay and colleagues, and Stuijfzand and colleagues' theoretical frameworks [34, 35]; the codes were organised into broader categories applicable to each objective and overarching themes through which the data could be described and interpreted. The research supervisors with expertise in qualitative research methods independently reviewed the themes to ensure quality assurance and increase data reliability, with any data discrepancies being resolved through group discussion.

## Researcher reflexivity

The principal researcher/author is a Northern Irish medical student studying in NI; they recognise their position had the potential to influence analysis and results. The author understands each individual has a different perspective on their situation, decisions and actions; dependent on numerous interacting factors including class, gender, race, environment, experiences and privilege [40]. The authors' perspective on how COVID-19 affected their mental health as a medical student could have shaped data collection, result analysis and discussion. Moreover, the researcher was a colleague and/or friend of some participants; it cannot be determined if this negatively or positively affected participants' information disclosure during the interviews. To the best of the researcher's ability, they attempted to separate themselves from participants; regarding them as interviewees rather than colleagues and analysed the data from a non-judgemental and neutral perspective to limit researcher bias potential.

## 3. Results

15 interviews were carried out amongst eligible participants; five intercalating medical students, two 4th year students, three 5th year students, three FY1 doctors and three FY2 doctors. 8 interviews were completed in person and 7 via zoom due to participants' choice; average length being 30.4 minutes. The range in interview lengths was 21–41 minutes; with a standard deviation of 4.24. After completing 15 interviews until data saturation, it became obvious medical students' mental health was impacted by the COVID-19 pandemic, mirroring existing literature. Most common mental health consequences included increased stress, anxiety and depressive feelings. Participants were asked to discuss their experiences from March 2020; when the pandemic was officially announced in NI, until March 2022; when all restrictions had ceased. Two different COVID-19 lockdown stages are referred to; the 'first summer lockdown' refers to March 2020-September 2020. The 'second winter lockdown' refers to October 2020-March 2021. The analyses resulted in three major themes: 1) Factors affecting mental health during COVID-19 and the consequences, and 2) Barriers to support service utilisation, and 3) Secondary impact on medical education progress. Various subthemes are addressed in the following paragraphs.

### 3.1 The factors affecting, and consequences on mental health during COVID-19

The majority of participants reported the pandemic to have negatively impacted their mental health, albeit with varying degrees of severity and for diverse reasons. Increased stress led to feelings of anxiety, depression and loneliness. Participants discussed various triggering and relieving factors contributing to these psychological consequences during the pandemic, applicable to the three subthemes; social, organisational and individual and/or psychological factors. The themes, subthemes and codes applicable to objectives 2 and 4 are demonstrated in Fig 1. These factors, their associated wellbeing consequences and participant quotations will be discussed in further detail in the following sub-sections.

**Social factors.** This subtheme relates to factors involving participants' social circle perceived to have impacted mental health during the pandemic. Many participants reported factors related to their living situation which negatively impacted well-being through increased feelings of isolation and loneliness; including living alone, far from friends or far from Belfast. Moreover, lack of in-person social support from colleagues, friends and partners was reported as a major trigger for mental health difficulties. Participant 2 spent the first lockdown in a separate bubble from their long-term partner, contributing to loneliness and low mood.

> *'I was impacted quite a lot because I do live far from Belfast, like two hours away. It's not as easy to just go meet someone, a lot of my friends are far away.' (8)*

> *'When you're around friends, everyone helps each other, there is support, (...) during COVID, your friends were a huge support lost.' (8)*

Factors related to participants' social life which negatively impacted mental health included lack of normal social opportunities; through university, sport or other extracurricular activities, and lack of ability to de-stress following difficult periods. Various participants described medicine as a very challenging degree, however justified by its sociability and comradery. COVID-19 removed the sociability, leaving behind only the most difficult aspects, contributing participants' feelings of disappointment, low mood and loneliness. Participant 11 discussed not feeling the same collectivism for the remainder of medicine, while participant 6 felt the pandemic made them more introverted.

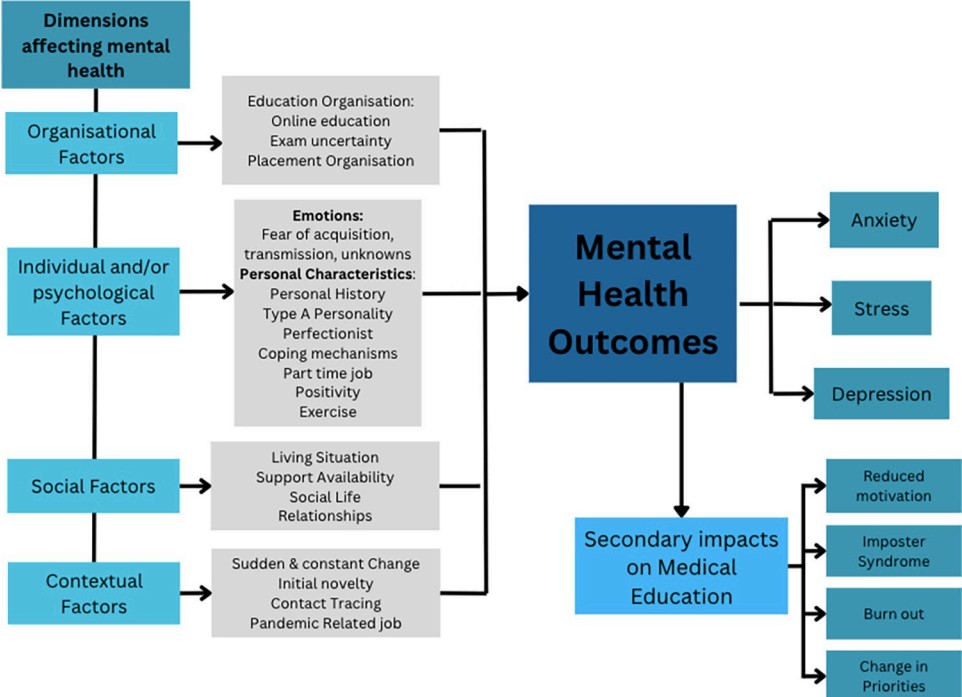

**Fig 1. Theoretical framework demonstrating the dimensions affecting mental health and the associated mental health outcomes, in addition to secondary impacts on medical education.**

*'That probably affected me most; the idea of medical school traditionally is that you're there to learn, but it's so much more than that, the social aspects of it are hugely important.' (15)*

*'Fourth year is definitely one of those years where I don't have many good memories.' 'There were a lot more downs than there were ups.' 'I definitely noticed a difference in like my mood, being more down, more lonely, more introverted.' (6)*

Upon returning to university in September 2020, many participants discussed the presence of a social stigma surrounding testing positive which negatively impacted mental health; through feelings of stress, guilt, fear and shame. Participants felt the social stigma was also secondary to being perceived as an increased transmission risk due to being a student who is also within healthcare environments. Participant 9 reported an additional level of stress as a medical student; fearing professional consequences if guidelines were broken.

*'There was quite a bit of stigma, you couldn't name who was positive and it was all this big shameful thing that you'd got COVID (. . .) I did kind of freak out'. (8)*

*'It was really stressful actually, looking back because I thought 'oh, if I'm found out that I'm here (at a party) and uni tell the GMC then I'm not going to be a doctor'. (9)*

Several participants mentioned the lack of ability to form new relationships during the pandemic negatively impacted their mental health. Due to online education, reduced social opportunities and placement changes; students could not meet new peers, leading to disappointment, anger and sadness. Particularly pre-pandemic intercalators (participants 10 and 11) felt the pandemic impaired their ability to connect into their new year. Now qualified,

they reported never feeling fully integrated into their graduating cohort. Participants 2 and 8 also discussed relationship breakdowns secondary to disagreements over government guideline adherence, which caused increased stress and worry.

*'One thing I struggled with was meeting new people, I would say most of my friends were relationships I had pre-pandemic, I didn't form many new relationships which was disappointing. Because I was joining a new year, I didn't necessarily ever feel a proper part of it.' (10)*

*'It was really hard to navigate that relationship with friends about having one bubble, (. . .) some people didn't respect those rules and then other people did take it a lot more seriously' 'It led to a breakdown in a friendship because people had different views. Still, that's not a friendship that's been rebuilt'. (2)*

Contrastingly, participants also recognised various social factors which had a protective effect on their mental well-being. This involved relying on new ways of socialising to project well-being; meeting friends via zoom, outdoor walks or employment. Participants discussed living in a student house with friends and/or other medical students facilitated the formation of a close-knit support system.

*'We were almost a little bit detached from it, because we were living in student houses with our friends (. . .) we were still having craic and good times with our friends at home (. . .) we were sort of all in it together'. (14)*

All participants discussed beneficial support from their job, university, family and/or peers which positively influenced mental well-being. Appreciation for family support was most prevalent in the first lockdown, while peer support increased in the second lockdown, when participants returned to university. Having a part-time job was recognised as a major mental-health relieving factor across participants; it provided an opportunity to socialise without breaking government guidelines. Even participants 7 and 12, who had a hospital-based COVID-19 job, reported feeling very supported.

*'I really enjoyed being at home with my family, the first time the 5 of us had been at home in a few years. That was definitely a relieving factor. I think I struggled more when I returned to 2nd year of uni. I moved into a house with other medical students. I didn't have the same support system of my family' (4)*

*'Even though work was awful, and the pandemic was grim. On that ward there were some really bad days, but everyone was in the same position and supported everyone.' (7)*

**Individual and/or psychological factors.** Participants reported various individual and/or psychological factors affecting mental health. Pandemic-related fear was the most common trigger for well-being consequences, however the most common fear ignitor transitioned throughout the pandemic. Initially, the most prevalent fear was of viral acquisition; due unknown side-effects or being unable to return home (mentioned by participants studying outside NI: 6, 9, 10 and 11). Participants reported fearing the unknown, as vaccines had not been developed and little was known regarding viral consequences. As the pandemic progressed, acquisition fear subsided, and fear of viral transmission to vulnerable family/patients became increasingly prevalent. Participant 10 discussed their acquisition fear subsided due to being young, healthy and unlikely to fall extremely ill.

*'At that stage, no one was vaccinated, you still didn't really know, some of my friends got it and had bad side effects; hair falling out and other long COVID symptoms (. . .) I was nervous for myself' (1)*

*'You are very, very conscious that you have been in hospitals. I've been around vulnerable and sick people and I've been exposed to different bugs, if I go and see other vulnerable people, then I could be doing more harm than good (. . .) That was an anxious time.' (6)*

Various participants discussed fear of COVID-19 acquisition during exam time, particularly final year participants 2, 5 and 15. Their fears of being too ill to study, or unable to sit the exam due to isolation caused increased stress, anxiety and disrupted sleep patterns.

*'I was scared of getting COVID-19, particularly around final year exams as there were implications for missing placement or missing exams. There was the fear of getting behind or not being able to sit the exams, definitely a few sleepless nights and heart pounding isolation fears surrounding contact tracing.' (5)*

The lockdown suddenly removed various mechanisms students would normally utilise to protect mental well-being; including social and sporting events. This was particularly experienced by 4th year students (2,5,15); their elective was suddenly cancelled, leading to disappointment, boredom and anger. Particularly stressful exam periods were further complicated by the lack of usual coping mechanisms; participant 8 described becoming obsessed with studying because there was nothing else to divert their attention, which negatively affected mental health.

*'Suddenly this big shift to a complete change of routine and how we knew how to do things was stressful, you didn't know what to expect.' 'You suddenly shifted to not being able to see anyone, you couldn't do the things you normally did, it was easy for that to affect your mood' (6)*

*'There was nothing else to focus on and you didn't have the same stress relievers, I think my obsession over giving 100% increased' (8)*

Furthermore, participants discussed various personal characteristics they felt exacerbated the COVID-19 effect on their mental well-being; having a prior mental health condition, or a type A and/or perfectionist personality. Participant 4 had experienced episodes of obsessive-compulsive disorder (OCD) and eating disorder behaviours years prior to the pandemic, they felt their mental well-being during COVID-19 plummeted, which had a secondary negative impact on their relationships and education. Similarly, participant 13 had a depression history pre-pandemic, they felt COVID-19 initiated a relapse; particularly in the second winter lockdown when the novelty was gone, weather had declined, and days were shorter. They felt the main exacerbating factors were online education, lack of social opportunities and the perceived medicine toxic-work culture which led them to avoid facing their mental health difficulties.

*'I became very obsessive with the structure. OCD traits that I had before the pandemic were definitely increased.' 'Because everything was so uncontrollable, I personally became obsessive over things that I could control, to the extremes that it negatively affected my mental health.' (4)*

*'Whenever 2nd year hit and the lockdowns happened, and you couldn't go out places, I think I definitely relapsed, I think it was even worse than the first time (. . .) it felt pretty rough' (13)*

*'there's a toxic work culture, especially within the NHS and within medicine, studying, there's so much to do. It's easy to convince yourself that it's normal to spend that much time just working towards some goal. It wasn't, I was doing too much because I was trying to artificially keep myself busy.' (13)*

Numerous participants reported having type A and perfectionistic personalities as traits that negatively affected the ability to cope with pandemic-imposed changes; including the lack of routine, purpose or control during lockdown.

*'we're all very much perfectionists. (. . .) we like to follow rules, and we like things by the book. And I think probably everyone as a whole was probably more likely to struggle than like Joe Bloggs on the street because we are so used to structure and nearly being stressed all the time (. . .) we put ourselves under more pressure to be busy'. (12)*

On the other hand, participants reported various individual and/or personal mental-health protective factors. A major coping mechanism discussed was creating a routine through studying, exercise or part-time jobs, this allowed participants to create a sense of control during the pandemic. Numerous participants felt despite the increased COVID-19 acquisition risk, part-time employment was essential to protect their mental well-being risk as it provided a legal opportunity to leave their house, gain experience, meet people, as well as public appreciation and pride.

*'It was actually really good, to feel like you kind of had a purpose and a role (. . .) It also meant I wasn't seeing the same people all day every day. (. . .) That structure helped a lot. I honestly don't know where I would have been without it.' (12)*

*'People were very supportive of the NHS, you felt like you could give something back, although you're only a student, it felt like a privilege to still be in hospitals and be around sick people'. (6)*

Participant 7, who had a job on a COVID-19 respiratory ward reported a mental toll from being exposed to dying patients separated from relatives. However, they recognised the job rewards outweighed the risks.

*'I don't think any 3rd year thinks you are going to be exposed to so much death, particularly young people dying, families being upset. It definitely took a toll mentally (. . .) the days were difficult.' (7)*

The most significant relieving factor described by all participants was exercise, particularly in the first lockdown when the weather facilitated outdoor activities. Exercising allowed participants to try something new, create a routine, unwind and safely meet people outdoors.

*'Everyday, I'd go for a fairly long run for like an hour, initially it was actually really nice, because I hadn't ever done that, it was springtime, it was bright outside. I was able to spend more time outside and properly appreciate it' (13)*

**Organisational factors.**   Organisational factors are related to participants' environment in which they learn and work. Medical education changes including increased online learning and reduced clinical opportunities were mentioned by all participants as factors negatively affecting mental health. Participants felt online learning removed their university student

experience; making them feel isolated, lonely and bored. Participants described online classes as demotivating and difficult to maintain concentration, which negatively affected well-being and medical education progress. Participant 14 felt the self-directed aspect of online learning made them feel alone and unsupported, leading them to question studying medicine.

*'I find it quite hard to focus compared to being there in person (. . .) you don't get the same sense of working together as when you're meeting people'. 'It was a very lonely year at times, a lot of that is due to zoom.' (6)*

*'There was a combination of COVID, online teaching, placement and things like that in third year, which I struggled with. I was like, this is just not for me, I can't do this, I can't self-teach'. (14)*

Participant 13 particularly struggled with the sudden change to online education; they felt their opportunity to gain a true student experience was removed, leading to feelings of entrapment, disappointment and depression. This led to reduced motivation and concentration for studies. Additionally, many participants described feelings of unpreparedness and impostor syndrome upon returning to in-person education after the prolonged period of online learning.

*'Most of my memories of 2nd year, especially during the actual strict lockdowns, were just me, in my room, on my laptop (. . .) it didn't feel like medicine. It didn't even feel like a degree. It just felt like something to get over and done with' 'It felt like the worst thing in the world.' 'The entire concept of time had kind of blurred into meaninglessness once all of the learning had been cancelled.' (13)*

*'Going from online university in 2nd year and straight into placement in 3rd year, feeling a bit like you don't really know what you're doing at all (. . .) I think was much more exaggerated because you've done online exams for two years' (1)*

Additionally, participants discussed feeling unwanted and a burden to overworked staff on placement, which was demotivating. Participant 15 described the clinical-teaching suspensions contributed to feelings of unpreparedness, anxiety and fears of being an inadequate doctor.

*'Staff were very busy, so they didn't have time for 3 medical students walking up to the ward asking for teaching. (. . .) That definitely affected my motivation to go in. We were just gonna be told, sorry, we have no time for you today. Go home'. (13)*

*'There was kind of the running joke amongst doctors at the time that we were like the COVID generation that haven't had placement in two years. Proper placement. We hadn't really sat proper 4th year exams. So, we were going to be awful doctors (. . .) that was a real worry.' (15)*

Contrastingly, participants discussed various mental-health relieving factors related to university organisation. Participant 9 enjoyed online education; reporting it was handier and provided more free time, while participant 10 felt it improved teaching consistency.

*'Before I'd have to get a bus to my lectures. I kind of enjoyed like just like waking up five minutes before and like tuning in and then I could not tune in if I want if I wanted to' (9)*

*'Because of the inconsistency and effect of COVID they arranged a lot more zoom teaching, which was more standardised'. (10)*

### 3.2 Barriers to utilisation of support services

Participants discussed various barriers to utilisation of mental health support services in NI; these are organised into themes of social, individuals and/or personal, organisational and contextual barriers. The themes, subthemes and codes applicable to objective 3 are demonstrated in Fig 2.

**Social barriers.**   Various participants discussed mental health stigma amongst students and HCPs as a social barrier preventing support utilisation. Participants perceived this stigma to exist as medical students and HCPs are often idolised as perfectionist individuals who do not suffer from invisible mental health problems. Participants discussed stereotypes they perceive to exacerbate the stigma; the belief that students are too young to be struggling mentally, and the belief medical students should know how to treat psychological conditions, therefore should not struggle from them themselves.

*'I guess, traditionally, the stigma of not wanting to be labelled as having a mental health diagnosis (. . .) not wanting to be the person you're learning to treat (. . .) we sort of think of people with mental health problems or mental ill health, on a lesser scale than physical visible health problems.' (15)*

*'You think you're so young, at the age of 22 you shouldn't be dealing with mental health issues.' (8)*

Participants recognised the social stigma leads to suppression of struggles due to fear and embarrassment, therefore help is not sought after.

*'I'd say most people probably do feel sort of conscious of what others think, we're conscious of not showing that we're not coping (. . .) people would feel very sort of embarrassed about it'. (10)*

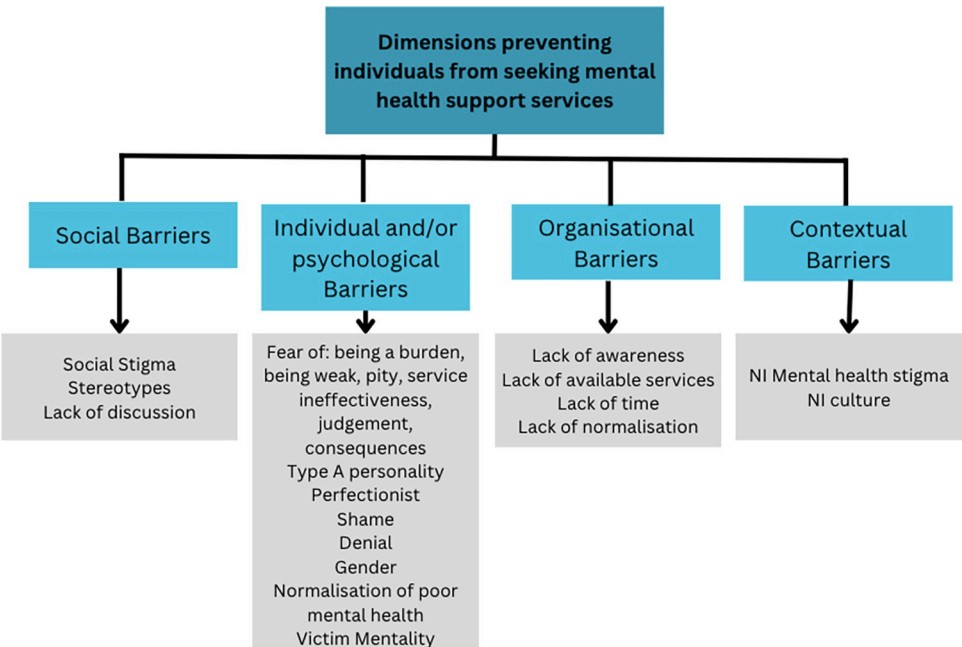

**Fig 2. Framework demonstrating dimensions of barriers preventing individuals from seeking mental health support services in NI.**

**Individual and/or psychological barriers.**   The most common perceived individual and/or psychological barrier preventing support utilisation was fear. Various types of fear were discussed; fear of being a burden, judgement, service ineffectiveness, being perceived as weak and unwanted professional or educational consequences. Participant 10 discussed the latter stems from being a perfectionist, and the stereotype medical students are always successful. Having a type A and/or perfectionist personality were commonly perceived personal barriers to support utilisation. Participants believed these traits make individuals stubborn, self-reliant and resilient, therefore more likely to persevere without help. Participant 4 discussed still feeling embarrassed and ashamed to have struggled mentally during the pandemic. Similarly, participant 8 and 13 admitted denial in recognising their struggles.

*'People don't want it to be a barrier to their medical education, just because that's something they're experiencing at this moment; they don't want to be a barrier that's there for life. Especially with medical education getting so interlinked with going into employment.' (1)*

*'There's a sort of perfectionistic tendency of medical students who think their entire life has to be perfect. (. . .) That's probably a barrier for people accessing help or admitting in the first place that they have something wrong.' (15)*

*'I think a major barrier would have been an embarrassment. Still to this day having recovered from those mental health issues, I still haven't spoken to any of my friends about it' (4)*

Numerous participants discussed normalisation of poor mental well-being during high-stress periods as a significant support utilisation barrier. Participants reported struggling during exam time; with increased stress, anxiety and disrupted sleep patterns. However, they accept mental health deterioration and persevere with the hope of improvement upon exam cessation.

*'Trying to address patterns that you've grown so used to as pathological, and as things that need fixed, it's hard' 'the idea of this work culture being normalised within medicine, meant for me, it took longer for me to realise what I was doing was unhealthy'. (13)*

**Organisational barriers.**   Participants discussed various organisational factors perceived to prevent mental health service utilisation; lack of knowledge of available services, service shortages and lack of time. Many participants felt the pandemic exacerbated these barriers due to the impersonal nature of online services, and service demand increased, but availability decreased.

*'I think busy placements, lots of placements, lots of exams crammed in, can make seeking mental health services and staying engaged with them difficult.' (15)*

*'Probably fear of online things. I think making the step to speak to someone about your mental health. It's quite a brave thing really, and probably quite a nerve-wracking thing to do face-to-face. But even more nerve wracking from your living room on a computer screen. It feels quite unnatural to be talking about something quite sensitive.' (2)*

Various participants discussed the need for normalisation of discussing mental health to protect mental well-being and encourage service utilisation when individuals first notice they are struggling, rather than when their condition reaches extremes.

*'We need to normalise that having mental health problems is okay and common. But you shouldn't accept it. It's not something that you should live with for your whole life.' 'I think if I knew about more people with mental health conditions, I know I would have been more inclined to share my struggles and even speak to people in a similar boat. But I just have no idea how I would do that.' (4)*

**Contextual barriers.**    Contextual barriers are related to participants' time and place in NI. When asked about support utilisation in NI, various participants with experience living elsewhere in the UK felt NI mental health stigma is higher, stemming from older generations which reduces likelihood of seeking support. Participants discussed because NI is small, it is likely you could encounter a familiar HCP, leading to embarrassment and fear of information spread.

*'it's such a small place and everyone knows each other. I mean, if things got really bad here and I had to go onto a psych ward, what are the possibilities that I bump into someone, or the doctor is someone's mum?' (9)*

*'There is a stigma, maybe not so from our generation, but certainly from older people, even including medical professionals (. . .) you hear offhand comments from like, maybe medical doctors on ward rounds, they'll say like, "oh, he has a whole host of mental conditions as well". (. . .) I think medical practitioners can say a lot of things off the backhand, (. . .) and it does have a derogatory tone associated with it.' (12)*

Participants labelled NI as a culture which does not sufficiently address or protect mental health, particularly amongst older generations who do not discuss sensitive topics. Participant 7 perceived NI society as not being aware of the spectrum of mental health conditions; minor problems often being exaggerated.

*'I think NI is backwards and we are trying to catch up with other places. (. . .) Here as a country, you keep your issues to yourself, and people don't share. But not just mental health problems, it's the same for lots of other things.' (7)*

'As a society in Northern Ireland, it's a lot more sweeping under the carpet, "let's not talk about it", or "I just have a headache", (. . .) less of actually "I'm exhausted, I'm burnt out, I feel depressed". And you know, "I have no empathy and no energy left to give". If that's really how you feel, maybe that's not a day you should actually be at work, or even as a student, you shouldn't be there. You know, you're not in the right position to learn.' (11)

*'I think a lot of people see it as a really bad thing and compare certain mental health things to extremes they see on TV and like people having psychotic breaks. It's not always like that, people can just have anxiety. I think people don't understand the spectrum of different mental health problems and psychotic conditions.' (7)*

However, it must be noted various participants also felt the COVID-19 pandemic shed a light on mental health and increased discussion. Therefore, participants discussed mental health awareness and stigma being a current but improving problem in NI.

### 3.3 The secondary impact on medical education progress

Several participants recognised the psychological effect of COVID-19 had an impact on their medical education progress, through burnout, and reduced motivation and concentration.

Participant 8 reported becoming obsessed with studying, which negatively impacted their motivation, concentration, and subsequently their learning. Depressive symptoms left participant 13 feeling increasingly tired with decreased energy for learning.

*"I wasn't having breaks and stuff then like I couldn't really concentrate. (. . . .) I was coming to the stage of burning out' 'I identified in myself lack of motivation, tiredness, not learning, not retaining information, I was just like, hitting a wall.' (8)*

*'I just felt very like tired and apathetic.' 'I didn't have much energy to study (. . .) even doing basic activities of daily living, felt more sluggish and a lot harder to do.' (13)*

Participant 4 felt the COVID-19 effect on their mental health negatively shifted their priorities, becoming obsessed with controlling other things.

*'My mental health affected my progress in medical education because pre-pandemic, my education would have always been my top priority. (. . .) I just did not have the same outlook. I always prioritised my diet and exercise and my 3rd-year placement definitely suffered. I didn't focus much on placement. I just wanted to leave and be in attendance for as short a time as possible.' (4)*

Moreover, various participants reported impostor syndrome feelings after online exams and classes; believing they did not deserve to study medicine and were not smart enough. This led to feelings of self-doubt and fear of being an inadequate future doctor. Participant 1 felt their impostor syndrome influenced their decision to take a year break from medicine, while participants 4, 8, 13 and 14 felt they lost their passion for medicine.

*'There was a massive point in third year when people did their first face-to-face exams (. . .) there was a bit of impostor syndrome of "God, we've never really done anything properly. Really, am I meant to be here?'. (1)*

*'I couldn't really say that I had much learning experience on placement. I maybe just lost the motivation or drive to actually like, want to do medicine' (8)*

Final year students and FY1/FY2 doctors discussed if the pandemic mental health effects influenced their decision to continue working in NI. Approximately half the participants discussed the pandemic highlighted the importance of an effective support system and felt they needed to stay close to home. Others discussed post-pandemic they felt trapped and isolated in NI, and in need of a change of scenery.

*'We were isolated in one geographical place, like everyone was for a long period of time' 'It was the idea that maybe there's something better and I'm not experiencing it. (. . .) It kind of definitely did influence my decision to leave.' (11)*

*'A part of me is weirdly thinking about like my mental health in F1 like, I don't really want to go away because how am I gonna cope with no family? No friends in a new city? With a really stressful job? I think my mental health would plummet'. (14)*

Participant 11 discussed their perceptions of higher Northern Irish mental health stigma and poor junior doctor well-being compared to other UK countries influenced their decision to work elsewhere.

*'From colloquially hearing some of the pressure or the additional pressures of being a foundation doctor in NI, (. . .) there's not a priority of well-being. If you have to stay late, that's just how it is. It's never actually like, "oh, there's a problem with the system when actually every F1 is staying late every day". (11)*

Contrastingly, various participants reported some present-day positive effects on medical education. Nearly all participants discussed their new-found appreciation for work-life balance. Participant 15 (FY2) felt this has continued into junior doctors' professional careers.

*'Coming up to exams (. . .) we'd have like sweated it out and like sat at the same table for so long, like, revising. Whereas now, I wouldn't do that. I feel like I would take breaks more, being nicer to myself (. . .) it's not really the be all and end all. I think that's how my priorities kind of did shift.'. (9)*

*'I think lots of people you talk to were going to go straight into training and sort of complete a training programme and rush through to become a consultant are now saying, "well, actually, I didn't get to do an elective. I didn't get to have fun during my F1 and F2 years. So, I'm gonna go and do an F3 (. . .) why would I not just enjoy myself when I can?"' (15)*

Furthermore, several participants recognised there were some positive pandemic mental health effects through developing resilience and coping mechanisms. Participant 7 felt the impact of COVID-19; although stressful at the time, has reduced long-term university stress.

*'I think we really started to learn, like, I need to do these things for myself so that I can function better.' 'I don't think I need to get like really, really stressed for uni anymore. There's more to life than sitting at a desk.' (7)*

*'As much as it was bad craic and I didn't enjoy it, I think I have changed for the better as a result of it. (. . .) I think it took getting that bad for me to actually seek help. So, then I went to therapy, and I built up like proper coping mechanisms' 'Now I still feel better overall'. (13)*

## 4. Discussion and conclusion

This research concluded the pandemic had a detrimental effect on Northern Irish medical students' mental health; common consequences being increased anxiety, stress, depression, as well as personal changes such as increased paranoia and obsessiveness. In addition, this study identified various triggering and relieving factors contributing to these consequences amongst students in NI which can be categorised into social, individual and/or psychological, organisational and contextual factors. Unanticipatedly, the research also identified some positive mental health consequences associated with the pandemic, such as building effective coping strategies, becoming more resilient and gaining an appreciation for an appropriate work-life balance. Furthermore, the study concluded the pandemic effects on mental wellbeing had a secondary negative impact on students' motivation in medical education. Finally, various barriers to utilisation of mental health support services amongst medical students in NI have been recognised which can fall under the same themes utilised above; including perfectionistic tendencies, social stigma, lack of awareness or fear of judgement.

### 4.1 Discussion

Previous literature completed across numerous countries identified the pandemic had a negative effect on medical students' mental health [7, 41–43]. As expected, this qualitative research

completed in NI concluded similar findings; common effects being increased stress, anxiety and depression. However, to the extent of the authors knowledge, this research was the first of its kind to identify pandemic mental health effects had a secondary negative impact on students' medical education progress; through reduced motivation, burnout and impostor syndrome. The imposter phenomenon experienced by various participants had further consequences; influencing a year out, burnout and questioning professional preparedness. Previous literature has identified impostor feelings are prevalent amongst HCPs and perfectionist type individuals. Additionally, literature concludes physician burnout correlates with poor work function, professional disengagement and low-quality patient care [44, 45]. This study identified perfectionist and Type A personality traits as common medical student traits, as well as negative predictors for pandemic-related mental health difficulties. It was evident participants possessing these characteristics struggled with the lack of routine or purpose imposed by the pandemic. Linking with previous literature, these study observations confirm the medical student cohort is at a higher risk of mental health difficulties and require high-level support, as such difficulties can have secondary consequences on education, future professional life and patient outcomes. It is imperative to increase medical student mental health awareness and support in NI; both in future pandemics, and day-to-day education.

Preceding literature identified various factors influencing medical students' and HCPs' mental health during the pandemic, similar to multiple factors identified in this research [34]. However, it was unexpected how different factors affect individuals in distinct mechanisms; the particular effect of a factor can vary, dependent on numerous variables including specific pandemic point, individual participant characteristics, medical education level, environment and social surroundings. For example, for students in early medical education years during the pandemic, main mental consequences were loneliness and isolation surrounding online education. Whereas for higher level students, main consequences were increased anxiety and stress surrounding balancing placement, COVID-19 acquisition risk and exams. This highlights the multifactorial nature of mental health and importance of intersectionality; multiple dynamic factors interact to affect each individual's mental well-being in different ways. Support services should be tailored to effectively match each individuals' needs, and adaptable to match the ever-evolving demand. Additionally, this research echoes previous literature which suggests patients with previous mental health disorders are at an increased risk of future conditions [46]; one participant struggled with depression reignition, and another with OCD behaviours during the pandemic. It is essential medical students with previous mental health history are identified and provided with additional support and encouragement to discuss their mental well-being and utilise services.

Most frequently reported psychological relieving factors during the pandemic included exercise, part-time job, work-life balance and strong support mechanisms. Interestingly, previous literature has identified part-time employment as a factor increasing stress amongst students [47]. Therefore, it was predicted having a COVID-19-related part-time job would exacerbate stress; due to increased acquisition risk and workload. However, all participants with a job reported it as a major relieving factor; providing social opportunities, clinical experience, and sense of structure and routine; two factors' participants struggled without during the pandemic [48]. In the future, these relieving factors should be increasingly utilised by medical schools to protect students' mental well-being, particularly in pandemics. Additionally, the 'MST' role; established due to the COVID-19 initiation is a valuable role which is synergistically beneficial to students and staff; it should remain and be established further cross other UK health trusts.

To the extent of the author and supervisors' knowledge, this was the first research carried out in NI assessing barriers mental health support utilisation amongst medical students;

common barriers included fear of consequences, appearing weak or stigmatisation. Participants discussed various barriers specifically applicable to the NI context; including the prevalent mental health stigma and mental well-being is not regarded as equivalent to physical well-being. Some participants perceived NI mental health stigma as higher than other UK countries, influencing their decision to leave NI for their professional life. The WHO has recognised the importance of tackling stigma, stating; 'the single most important barrier to overcome in the community is the stigma and associated discrimination towards persons suffering from mental and behavioural disorders' [49]. This study emphasises the need for an active mental health movement in NI to reduce stigmatisation and encourage individuals to seek required help. However, various participants reported feelings that mental health discussion has improved post-pandemic due to more individuals struggling mentally. This highlights there has been one step in the right direction, and the momentum needs continuing.

Various participants discussed a barrier preventing support utilisation is the acceptance of lower baseline mental health during stressful periods. Instead, individuals preserve and utilise the belief their well-being will improve upon cessation of the stressful period. These behaviours are maladaptive, they encourage participants to not take breaks or steps to improve mental well-being [50]. NHS junior doctors' function in a similar manner; it is often accepted they will be under extreme pressure. They have been indoctrinated it is the social norm to have poor mental health due to high workplace stress and anxiety, causing acceptance of unfavourable working conditions [35]. This has led to numerous junior doctors experiencing burnout, contributing to the NHS workforce crisis. 2019 data demonstrated only 35% of junior doctors choose to immediately continue their training after their two foundation training years; many opting to take a career break, others leaving the profession [51]. This research identifies normalisation of low mental well-being begins from medical school. In order to make a change amongst junior doctors, mental health prevention and protection must begin in undergraduate training. In the future, careful steps must be taken to normalise mental health as a common problem, without encouraging students to accept experiencing poor mental health themselves.

It was unanticipated some students reflect on the pandemic positively; believing it improved resilience and mental-health coping mechanisms. Other perceived benefits included enhanced clinical experience through healthcare jobs and newfound work-life balance appreciation. Medical students often have to juggle various activities; studying is frequently prioritised while self-care and leisure time are disregarded [52]. Literature demonstrates work-life balance is an integral component to doctors' mental well-being; the UK General Medical Council (GMC) advertises its importance for self-care, ability to cope with lifelong learning and deliver high-quality patient care [52, 53]. The GMC recognises poor work-life balance amongst junior doctors leads to exhaustion and low retention rates [45]. However, this research identified various students experienced a shift in priorities during the pandemic; learning the importance of mental well-being protection. This is a beneficial outcome; if it sustains long-term it has the potential to improve future students' mental health. It is important to encourage and maintain these current attitudes and increase education on the importance of work-life balance.

## 4.2 Strengths and limitations

As far as the author is aware, this research provides a new perspective not yet explored by existing research. There are high levels of participant variation; including various medical education levels, ages, genders and individuals from NI who studied elsewhere. The principal researcher independently carried out all 15 interviews, data anonymization, storage and analysis; ensuring high levels of participant confidentiality and data protection.

Due to the study qualitative nature, and all participants being Northern Irish white ethnicity, result generalisability and cross-population transferability may be limited. However, the results could be utilised to guide future research studies exploring factors influencing mental health and their consequences. Replicating this research in other contexts would give the opportunity to compare the strength of the study results. Additionally, there were 12 female and 3 male participants due to a higher proportion of females in NI medical education; therefore, this study cannot compare COVID-19 mental health effects on different genders. In addition, purposeful and snowball sampling provides the potential for recruitment and response bias; this sample may not be representative all medical students' experiences across NI. However, various link individuals were utilised to increase cohort diversity and mitigate this risk. The principal researcher is a NI medical student; creating the potential for researcher bias. In order to mitigate this, the bracketing approach was adopted; this involves putting aside one's own beliefs about the phenomena consideration to prevent biasing their observations [54]. Moreover, individuals with active mental health conditions were excluded in order to avoid exacerbation of mental ill health; this created the potential for exclusion of individuals' perspectives who experienced significant and continuing negative pandemic mental health effects. Finally, there was a 3-year gap between the initiation of the pandemic in NI and the interview period; it is possible confounding factors could have affected participants mental health within this period. The author attempted to mitigate this risk by focusing the interview discussion solely on factors influenced by COVID-19 and limiting discussion to between March 2020 and March 2022.

## 4.3 Recommendations

Key factors positively influencing participants' mental health included exercise, work-life balance, part-time job and effective support system. NI medical schools should maximise these positive influences to protect students' mental well-being, while mitigating identified compromising factors. Additionally, a significant effort is required to reduce NI mental health stigma; potentially through increased mental health openness, support availability and awareness. Education on the importance of mental health protection needs to be incorporated into school-level education, to normalise mental health as an integral well-being component. Due to the small scale of this research, further qualitative and quantitative research is required amongst a larger variation of individuals to increase result generalisability. Additional research could be carried out specifically comparing the COVID-19 mental health effect on different genders, ages or socioeconomic status, to further expand perspectives. Furthermore, additional research is required into the long-term pandemic effects on students' mental health.

## 4.4 Conclusion

The COVID-19 pandemic has significantly impacted medical students' mental health; exacerbating existing stressors and introducing new challenges. The abrupt shift to online learning and clinical restrictions has disrupted students' traditional educational experience, leading to increased stress, loneliness and depression. This mental health impact had a secondary negative impact on medical education progress during the pandemic; however, reflecting on the pandemic in the present-day, there were some positive effects through improved resilience, work-life balance and appreciation for studying medicine. This research emphasises medical students' mental health, in addition to support utilisation, as topics requiring attention in NI. There is a growing demand for more discussions about mental health to recognise it as a crucial aspect of overall well-being. This increased dialogue could help reduce stigma and encourage help seeking behaviours amongst individuals. It is crucial to prioritise students' mental

health during challenging periods by providing adequate and accessible support to help them navigate unprecedented circumstances and cultivate resilience.

## Acknowledgments

The authors thank all the participants for sharing their experiences.

## Author Contributions

**Conceptualization:** Claire Whiteside, Gonnie Klabbers.

**Data curation:** Claire Whiteside, Gonnie Klabbers.

**Formal analysis:** Claire Whiteside, Gonnie Klabbers.

**Investigation:** Claire Whiteside, Gonnie Klabbers.

**Methodology:** Claire Whiteside, Gonnie Klabbers.

**Project administration:** Claire Whiteside, Gonnie Klabbers.

**Resources:** Claire Whiteside, Gonnie Klabbers.

**Software:** Claire Whiteside, Gonnie Klabbers.

**Supervision:** Gonnie Klabbers.

**Validation:** Claire Whiteside, Gonnie Klabbers.

**Visualization:** Claire Whiteside, Gonnie Klabbers.

**Writing – original draft:** Claire Whiteside, Gonnie Klabbers.

**Writing – review & editing:** Claire Whiteside, Gonnie Klabbers.

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
