## [Decision Letter · Decision Letter 0]

2 May 2024

PONE-D-24-04972Exploring the perceptions of the effect of the COVID-19 pandemic on the mental well-being of medical students in Northern Ireland; A Qualitative StudyPLOS ONE

Dear Dr. Whiteside,

Thank you for submitting your manuscript to PLOS ONE. After careful consideration, we feel that it has merit but does not fully meet PLOS ONE’s publication criteria as it currently stands. Therefore, we invite you to submit a revised version of the manuscript that addresses the points raised during the review process.

We look forward to receiving your revised manuscript.

Kind regards,

Md Nazmul Huda, PhD, MSS, BSS

Academic Editor

PLOS ONE

Journal Requirements:

**Additional Editor Comments:**

In addition to reviewers' comments, please address my below comments:

1) Please indicate the study design in the abstract. Did you follow a descriptive design? Or any other study design? Indicate the analysis technique in the abstract.

2. In the abstract, results lack substances. For example, you talked about impact without indicating the impacts of covid. Please provide more details of impact for the readers.

3. Reduce the number of objectives and mention the objective you addressed and indicated in the paper title/abstract. Or change the title. This means there should be consistency between your paper aim, title and results you described. The current title only indicates the impact. But you described barriers etc. Make necessary changes everywhere.

4. Analysis technique is absent. Please detail.

5. Please summarise your findings in the discussion section before comparing them with other studies.

Reviewers' comments:

Reviewer's Responses to Questions

**Comments to the Author**

1. Is the manuscript technically sound, and do the data support the conclusions?

Reviewer #1: Yes

Reviewer #2: Yes

Reviewer #3: Yes

2. Has the statistical analysis been performed appropriately and rigorously? 

Reviewer #1: Yes

Reviewer #2: No

Reviewer #3: No

3. Have the authors made all data underlying the findings in their manuscript fully available?

Reviewer #1: Yes

Reviewer #2: No

Reviewer #3: Yes

4. Is the manuscript presented in an intelligible fashion and written in standard English?

Reviewer #1: Yes

Reviewer #2: Yes

Reviewer #3: Yes

5. Review Comments to the Author

Reviewer #1: This is a qualitative study article on the effect of COVID-19 on the mental well-being of medical students in Northern Ireland. The discussion on the basics is fine and based on a broad understanding of the field. However, it seems to need to with a few sentences in the introduction by focusing more on Northern Ireland during the COVID-19 time to distinguish it from other similar studies (Prevalence rate, government services, accessibility to vaccine...). Also, let me know why the authors decided to interview participants three years after COVID-19. How are authors sure in these years participants were not involved with other emotional damage?

Reviewer #2: Thank you for the paper

- Abstract needs revision. I do not know how the authors mentioned significant results in qualitative. This section needs revision - Methods part needs comprehensive revision

- Add rigor in Method

- In results, show themes, subthemes and code

- Add demographic chchs of the study sample

- Revise discussion and link to results

Reviewer #3: 1. I think, your paper is well written but it is more suitable for PLOS Mental Health Journal.

2. Line number 135: "The principal researcher/author is a female Northern Irish medical student studying..."

why is it necessary to mention the gender of the principal researcher/author?

3. Line number 151: "... average length being 30.4 minutes."

I think Standard deviation should be included here in order to understand the dispersion/spread of the interview length.

4. Line number 593: "This was the first research carried out in NI assessing barriers mental health support utilisation amongst...."

If you are 100% sure about what has been written, then it is alright... Congratulation on your work. But if not, please write as follows: "as far as we know" or "to our knowledge". In that case, it would not create any possible conflict of interest in future.

5. Is NHS under the jurisdiction of Northern Ireland? or it is HSC that is under the jurisdiction of NI? I am pointing out it as the UK NHS has been mentioned in Line number 105. NI may have different publicly governed/funded healthcare system if I am not wrong.

6. PLOS authors have the option to publish the peer review history of their article (what does this mean?). If published, this will include your full peer review and any attached files.

Reviewer #1: No

Reviewer #2: **Yes: **Abd Alhadi Hasan

Reviewer #3: No

---

## [Author Response · Author response to Decision Letter 0]

8 Oct 2024

Responses to comments made by the academic editor: 

I have checked the PLOS style requirements and ensured the manuscript; including tables, figures and references meet the required styles.

This is a qualitative research study. Therefore, all data is in the form of recordings which contain identifiable and sensitive information. Apart from length of participant interviews which is included in the revised manuscript, there is no numerical data or graphs. There are anonymous quotations within the manuscript results section. Upon participant recruitment, the following information was in the participant information letter:

‘The data will be stored anonymously and be inaccessible to all excluding the principal researcher and supervising researcher. The data will be used to write up a thesis project which will be submitted to journals to be potentially published. If the opportunity arises, the results of the research will be made into a poster or PowerPoint for presentation for the potential to be presented at Maastricht University and or a national/international conference. If desired, the thesis will be made accessible to participants after completion. participants were informed all data’.

Upon applying for ethical approval from the Maastricht University Ethics committee, the following information was in the approval request letter:

‘Identifiable data, such as audio recordings and transcripts will only be accessible by the principal researcher. Anonymous data will only be accessible by the research team; made up of the principal researcher and two research supervisors. The principal researcher will act as the data controller.’

‘Other individuals will not be able to access the data as it will be sorted on a password protected file on a password protected laptop during the project. After the project it will be stored on the UM server in password protected files. The data will not be printed.’

‘The data will not be shared or used in future research. However, the thesis has the potential to be published in a journal. Therefore, the publication could potentially be reviewed in a literature review. Potential publication will be referenced in the information and consent form.’

Therefore, sharing the data would both violate the agreement with the participants prior to data collection, in addition to the agreement made upon approval for the research with the Maastricht University ethics committee. The data is currently being securely stored in encrypted files by the principal researcher. Anonymous transcripts can be obtained from the principal researcher, Claire Whiteside on request but they cannot be shared publicly. The University of Maastricht Ethics committee address is Universiteitssingel 40, 6229 ER Maastricht, Netherlands and contact email is ethical_clearance_gh@maastrichtuniversity.nl

The methods section in the abstract has been extended to include additional study design information and analysis technique.

The results section of the abstract has been extended to include additional information. 

The title has been changed to more appropriately encompass the four study objectives. There is now consistency throughout the manuscript. 

Section 3; methods has been expanded to include more information regarding study design, participant demographics and data analysis technique. 

An additional paragraph has been added in section 5 to summarise the research findings prior to comparing the results with other studies. 

Responses to comments made by reviewers:

Reviewer 1: 

Section 1; the introduction has been adjusted to include more information relevant to COVID-19 in Northern Ireland, such as prevalence, death rates and vaccine uptake.

The participants were interviewed in June 2023. The COVID-19 pandemic initiated in Northern Ireland in March 2020 and all restrictions has ceased 2 years later in March 2022. Therefore, there was a 15-month gap between complete cessation of the pandemic in NI and interviewing participants. The principal researcher decided to complete the research at this point because this is when they were completing a masters in Global Health at Maastricht University. The research was carried out for the purpose of their master’s thesis. The principal researcher felt this was an appropriate time to complete this retrospective qualitative research as participants would be able to recall the effects of the pandemic on their mental health during the pandemic, as well as perceived secondary effects on their medical education; both during the pandemic and at the time of data collection. 

Reviewer 2: 

The abstract has been revised, with the word ‘significant’ removed and more detail in method section. The results section has been changed, with added adjusted theoretical frameworks (figure 1 and 2) to demonstrate themes, sub-themes and code. Table 1 has been added to demonstrate the demographics of the study sample. The discussion has been revised with added summary paragraph. 

Reviewer 3:

I will consider applying to the PLOS mental health journal if the manuscript is not successful in PLOS One. 

The gender of the principal researcher has been removed from "The principal researcher/author is a female Northern Irish medical student studying..."

Standard deviation of length of interviews has been added and interview lengths are demonstrated in table 1. 

Necessary adjustments have been made to the text to reduce opportunity for possible future conflicts of interest. 

The NHS is the National Health service of the UK; including Northern Ireland. There are different branches of the UK health system. NHS England is centrally funded from the Department of Health and Social care while Northern Ireland receives a grant from the UK treasury which funds the Department of health, social services and public safety for Northern Ireland (DHSSPS). The slight difference between health systems in England and NI is that both the health and social care are provided through an integrated service. Therefore, Health and social care Northern Ireland (HSCNI) is part of the overall UK NHS. The sentence (now line 155) has been adjusted to more appropriately reflect this.

---

## [Editor Report · Decision Letter 1]

6 Nov 2024

Exploring the perceptions of the effect of the COVID-19 pandemic on the mental well-being and medical education of medical students in Northern Ireland, in addition to the perceived barriers to seeking support; A Qualitative study

PONE-D-24-04972R1

Dear Dr. Claire Whiteside,

We’re pleased to inform you that your manuscript has been judged scientifically suitable for publication and will be formally accepted for publication once it meets all outstanding technical requirements.

Kind regards,

Md Nazmul Huda, PhD, MRes, BSS

Academic Editor

PLOS ONE
---

## [Editor Report · Acceptance letter]

11 Nov 2024

PONE-D-24-04972R1 

PLOS ONE

Dear Dr. Whiteside, 

I'm pleased to inform you that your manuscript has been deemed suitable for publication in PLOS ONE. Congratulations! Your manuscript is now being handed over to our production team.

Kind regards, 

on behalf of

Dr. Md Nazmul Huda 

Academic Editor

PLOS ONE